# Simulating Future Land Use and Cover of a Mediterranean Mountainous Area: The Effect of Socioeconomic Demands and Climatic Changes

Diogenis A. Kiziridis [1], Anna Mastrogianni [1], Magdalini Pleniou [2], Spyros Tsiftsis [3], Fotios Xystrakis [2,*] and Ioannis Tsiripidis [1]

1 Department of Botany, School of Biology, Aristotle University of Thessaloniki, 54124 Thessaloniki, Greece
2 Forest Research Institute, Hellenic Agricultural Organization "DIMITRA", 57006 Vassilika, Greece
3 Department of Forest and Natural Environment Sciences, International Hellenic University, 1st km Drama-Mikrochori, 66132 Drama, Greece
* Correspondence: fotios.xystrakis@fri.gr

**Abstract:** Land use and cover (LUC) of southern European mountains is dramatically changing, mainly due to observed socioeconomic demands and climatic changes. It is therefore important to understand LUC changes to accurately predict future landscapes and their threats. Simulation models of LUC change are ideal for this task because they allow the in silico experimentation under different socioeconomic and climatic scenarios. In the present study, we employed the trans-CLUE-S model, to predict for 2055 the LUC of a typical southern European sub-mountainous area, which has experienced widespread abandonment until recently. Four demand scenarios were tested, and under each demand scenario, we compared three climatic scenarios, ranging from less to more warm and dry conditions. We found that farmland declined from 3.2% of the landscape in 2015 to 0.4% in 2055 under the business-as-usual demand scenario, whereas forest further increased from 62.6% to 79%. For any demand scenario, differences in LUC between maps predicted under different climatic scenarios constituted less than 10% of the landscape. In the less than 10% that differed, mainly farmland and forest shifted to higher elevation under a warmer and drier climate, whereas grassland and scrubland to lower. Such insights by modelling analyses like the present study's can improve the planning and implementation of management and restoration policies which will attempt to conserve ecosystem services and mitigate the negative effects of socioeconomic and climatic changes in the mountainous regions of southern Europe.

**Keywords:** cropland; pastureland; shrubland; woody encroachment; rewilding; transhumance; livestock; land use and land cover change; LUCC simulation; projections

## 1. Introduction

Low-intensity farming and traditional management practices have shaped Europe's Mediterranean mountains for centuries [1]. Nevertheless, the abandonment of these marginalised areas has been pervasive after World War II, with significant consequences on the rural landscapes [2]. Be they positive or negative, the consequences of land abandonment on culture, biodiversity and ecosystem functioning depend on the characteristics of an area [3]. On the one hand, for example, land abandonment can constitute a threat due to the closing of the landscape, with the domination of scrub and forest species replacing endemic and rare species of the previously open habitats [4]. On the other hand, the scrub and forest species can have a positive impact to an area with history of overexploitation [5]. A main driver of abandonment is the economic marginalisation of these areas due to unfavourable socioeconomic and institutional conditions, such as a country's uneven distribution of income between cities and countryside, and of subsidies favouring the intensive farming in the lowlands versus the extensive farming in the uplands [6].

Additionally, the climate of the mountainous areas, together with their remoteness and topography, renders them the first to be abandoned from the countryside under such unfavourable socioeconomic conditions, because these characteristics further limit agricultural profitability and competitiveness in the local, national and international markets [2,3]. The elevational ruggedness of the mountainous landscape leads to climatic conditions of greater agricultural challenges due to the higher spatial, daily and seasonal variability in temperature and precipitation in comparison to the lowlands [7]. Moreover, climate change at the Mediterranean mountains is predicted to be more dramatic than in other European mountains by year 2055, due to a greater increase in temperature, and a greater decrease in precipitation annually but also during spring with the start of the vegetative period [8]. These dramatic changes are expected to affect the land's suitability for typical crops [9], forest species [10], habitats [11], and irrigation [12].

Thus, there is a need to accurately predict the future land use and cover (LUC) of mountainous areas of the Mediterranean in relation to climate change and its interaction with the local topographic and socioeconomic conditions [13]. This can facilitate the development of measures for the mitigation and adaptation against forthcoming threats. Adaptation is an important factor shaping the future severity of climatic and socioeconomic impacts on agriculture and biodiversity [14]. A key question is whether LUC change will be more affected by changes in the socioeconomic or the climatic conditions of the mountainous areas. The consequent prioritisation of socioeconomic or climate-related policies for ensuring food production and biodiversity conservation will hence require the processing of [9]: (1) biophysical and socioeconomic factors, and their relation to LUC change; (2) future LUC for evaluating its characteristics, and the possibilities for adaptation both in space and in time; and (3) the uncertainty related to future socioeconomic, climatic and LUC trajectories.

Spatial modelling of LUC change can disentangle the effects and uncertainties of these factors via the in silico experimentation of different climatic scenarios under assumed socioeconomic demands for various types of LUC [15,16], as we attempt in the present study. The modelling of LUC has proved a reliable tool for investigating such scenario projections of LUC change for the future [17]. In particular, modelling with scenario projections for the investigation of future LUC is important because it enables the more transparent comparison of methodologies from different researchers and for different areas [18]. Additionally, it provides information for the more efficient application of mitigation and management measures. For example, measures can be more targeted and cost-effective by knowing where in the landscape forest expansion is more related to land abandonment than to climate change, together with their correspondingly different species composition and time scale of expansion [19].

In the present study, we investigated the effects of climate change on the prediction of LUC in year 2055 for a Greek sub-mountainous area representative of Mediterranean areas with history of abandonment [20]. We assumed different scenarios of demand in transitions of LUC types. For each demand scenario, we compared the LUC predictions under three climatic scenarios, from more to less optimistic one, to find: (1) how much the demanded LUC transitions and composition differed between climatic scenarios; (2) how much the predicted maps differed in the allocation of demanded LUC; and (3) what were the environmental characteristics of the allocation differences between climatic scenarios. Climate was expected to have a strong effect on the predictions of future LUC, given the anticipated dramatic changes in the mountainous climate of the Mediterranean for the middle of the 21st century [8]. We further discuss our results in the context of future socioeconomic demands and climatic changes in the region.

## 2. Materials and Methods

### 2.1. Study Area and Historical Conditions

The study area was composed by five circular sites of 6 km in diameter each, with a total cover of 141.4 km$^2$, located in the Pindus mountainous region of Greece (Figure 1).

We chose this study area as a system typical of the Mediterranean mountains, because the region historically had extensive farmlands and grasslands exploited traditionally by transhumance and low-intensity farming until the 1940s, but land abandonment later on led to dramatic changes in the landscape [20,21]. We decided to work with circular sites because they have the lowest ratio of perimeter-to-area [22,23], and they would hence be less sensitive to edge effects in vegetation sampling we did for a different study. We chose five such sites for maximising the heterogeneity of the samples in relation to environmental conditions and vegetation diversity, as well as to abandonment.

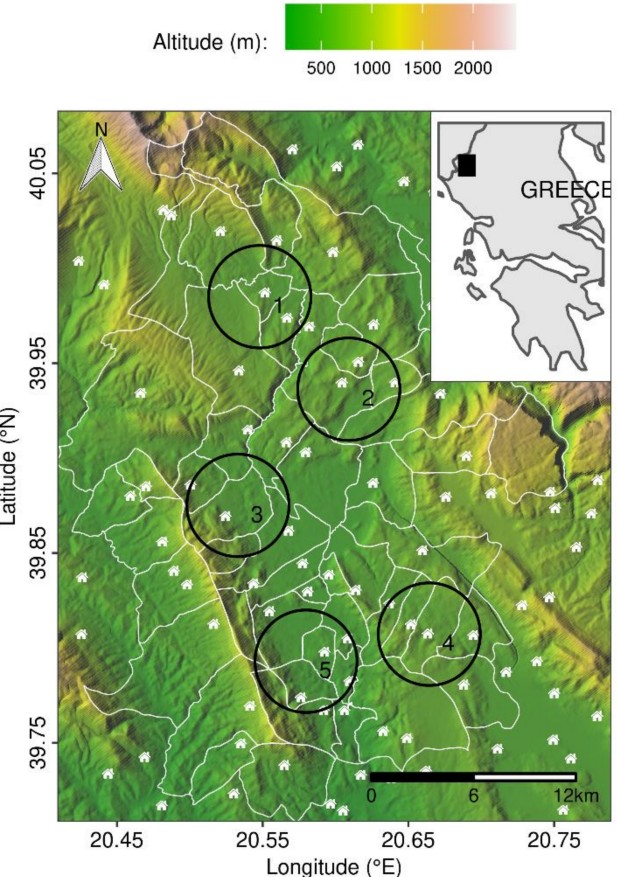

**Figure 1.** The five circles constituting the study area's sites. The inset's black-filled rectangle shows them in the region. White lines indicate boundaries of local municipal districts, and white symbols are for settlements.

The region's woody taxa categorise it to the "thermophilous deciduous oaks" vegetation formation [24]. In a previous study, we mapped the area's LUC for years 1945, 1970, 1996 and 2015 [20]. The mapping focused on five LUC types which are broad steps of progressive vegetation succession: farmland, grassland, open-scrub, closed-scrub and forest. Settlements and bodies of water were excluded, leading to a mapped 138.4 km$^2$ cover, which is smaller than the circles' total cover. Visual interpretation of orthoimages was used to identify LUC, before proceeding to vectorisation, and rasterisation at 25 m resolution. The biophysical and socioeconomic conditions of the study area were downscaled to the same resolution.

The biophysical and socioeconomic conditions of the study area concern the 1945–2015 period (Figure S1 in Supplementary Materials). The elevation ranges from 248 to 1203 m, and the slope ranges from 0–48°. The region's climate is the "Csa" hot-summer Mediterranean type according to the Köppen–Geiger scheme of classification [25]. The region has a history of low-intensity farming until the 1940s, but abandonment of farmlands and grasslands has commonly led to vegetation succession and afforestation [20,21]. During

the 1996–2015 period, which was used for the projection of the demand scenarios, the population density median decreased from 17.3 to 12.1 inhabitants km$^{-2}$, and the livestock density median decreased from 135.9 to 51.9 small grazing livestock units km$^{-2}$. Data for the geological substrates were taken from a previously documented map [26]. Data for the population and livestock sizes were from national censuses, and retrieved from their official online sources [27,28]. More details about the LUC mapping, and the conditions of the study area, can be found in the relevant work published previously [20].

### 2.2. Projections of Climatic and Socioeconomic Conditions to the Future

The target horizon in the future for our predictions was year 2055. Projections of socioeconomic and climatic conditions were made for suitability predictors which varied in time from our dataset of predictors (Figure S1): the 19 bioclimatic variables, population and livestock densities, and settlement proximity. No projections were made for the fixed in time predictors: elevation, slope, northness, eastness, and presence of four types of bedrock.

### 2.2.1. Climate

For climate in the future (Figure S2), the scenarios we employed were from the following Shared Socioeconomic Pathways (SSPs) in the 6th phase of the global collaboration among climate-modelling institutions known as CMIP6 framework [29]: (1) sustainability, in the direction of a more sustainable path, stressing more equitable development that adheres to perceived environmental constraints (SSP126, herein called SSP-SUST); (2) regional rivalry, in the direction of a nation-centred path, worrying about competitiveness, security, and regional issues. (SSP370, herein called SSP-RIVAL); and (3) fossil-fuelled development, in the direction of quick technical advancement and the development of human capital (SSP585, herein called SSP-FUEL). Average monthly values of total precipitation, as well as of mean, maximum and minimum temperature were retrieved from the CHELSAfuture V2.1 dataset of 1 km resolution for the historical period of 1981–2010, and for the future projection period of 2041–2070 which has the target year 2055 in the middle [30]. For each of the three climatic scenarios of the future period, we averaged the monthly time series from the five models provided freely by the Swiss Federal Institute for Forest, Snow and Landscape Research, for each of the four meteorological variables [30]. Thus, for each combination of cell, meteorological variable and month, we had two values, which were that month's average for the 1981–2010 and for the 2041–2070 periods. We applied correction and downscaling of the future period's monthly averages with the R package "meteoland" [31]. The correction of the meteorological variables was based on the comparison between the monthly time series from the CHELSAfuture for the 1981–2010 period, and the reference CHELSAcruts monthly series for year 1996 [32], which was in the middle of the 1981–2010 period, and which was already used in our historical dataset. We used the CHELSAcruts dataset because despite being cruder due to its delta-change method by B-spline interpolation of anomalies, it extends back to year 1901, and hence covers our earliest year of 1945. The corrected monthly time series for the four meteorological variables were downscaled to 25 m resolution. Finally, we calculated for year 2055 the 19 bioclimatic variables of Worldclim from the downscaled monthly time series with the R package "dismo" [33].

### 2.2.2. Population

For population, we tried seven different projection models (Figure S3), and we finally selected the one with the best performance (Figure S4), according to the following validation procedure. The models were calibrated with historical census data from years 1911–1967, and then predicted the following 44 consecutive years until year 2011. Year 2011 was the latest year of census data and was used for validating the predictions by the models of the same year. We selected this 44-year prediction window because the best-performing model according to the validation would be employed to predict the population in each district 44 years after the latest census year of 2011, i.e., in year 2055. To enable the calculations, we

linearly interpolated any missing values in the historical time series. The first model was a naive one, i.e., it predicted the same value as the last calibration year for the projection years. The next three models were the automatic ARIMA, Exponential Smoothing, and Neural Networks with the default settings of the R package "forecast" [34]. The last three models were Logistic, possessing the two parameters of growth rate *r* (units of population change per year) and of carrying capacity *K* (units of population size). The Logistic models differed in the estimation of *r*, since *K* was identically estimated as the maximum population size observed in the 1911–1967 time series of each municipal district. In all three Logistic models, the parameter *r* was equal to the (*end–start*)/*end* value of population size at the start and end of a time interval, respectively, divided by the interval's duration in years.

For the first Logistic model (lm-based), we first had to fit a linear model to the 1911–1967 time series of population size, and *r* was calculated from the 1911 and 1967 predictions of population size by the linear model. For the second Logistic model (start–end), we directly used the raw 1911 and 1967 population sizes for *r*. For the last Logistic model (weighted), we divided the 1911–1967 period to five sub-intervals of approximately equal duration, calculated *r* for each sub-interval, and the final *r* was the weighted average of the *r* values of the five sub-intervals. The weighting was exponential, such that each previous sub-interval had its weight halved in the calculation of the average.

Model performance was quantified with the metric "mean Absolute Scaled Error", due to its better properties and behaviour [35]. This metric is the ratio of two errors. The numerator holds the absolute error of a projection, e.g., the absolute difference between the 2011 projection and the census value of 2011 in our validation case. The denominator holds the average absolute error of naively predicting the same value in the calibration time series, and for the duration of the prediction window. In the case of our validation with the 44-year window, the calculations used the 1911–1967 calibration period, starting from predicting the same value in 1955 as in 1911, until predicting from 1923 the same value for 1967, and subsequently taking the average absolute error of the naive predictions versus the corresponding actual values from 1955 to 1967. A mean Absolute Scaled Error above one indicates an absolute error which is larger than the average error of naively predicting the same value after 44 years inside the calibration period, averaged over all possible 44-year predictions in the 1911–1967 calibration period.

The weighted Logistic model exhibited the best performance and was hence employed to predict the 2055 population size in each municipal district (Figure S5). For the projection from 2011 to the future, we calibrated with data from all available historical data from the 1911–2011 period (weighted average *r* from nine sub-intervals).

### 2.2.3. Livestock and Settlement Proximity

For livestock, we followed the same procedure as with population size, but the census data were from the 1961–2021 period (Figure S6). The models were hence calibrated with census data from years 1961–1987, and then predicted the following 34 consecutive years until year 2021. The validations and comparisons showed that the weighted Logistic model was not worse than others, and was hence selected to have the same model as for population size (Figure S7). Parameterising again the weighted Logistic model from all available historical data, we predicted the number of small livestock units at the municipal districts in year 2055 (Figure S8).

For settlement proximity, we used the same values for 2055 as in the latest historical year 2015, since proximity change was minor during the 1945–2015 period (Figure S1A).

### *2.3. LUC Change Model*

The model of LUC change was validated against reference historical data, and then was called to predict the 2055 future LUC under different climatic scenarios to analyse the comparisons of climatic scenarios in mainly three different ways (Figure 2).

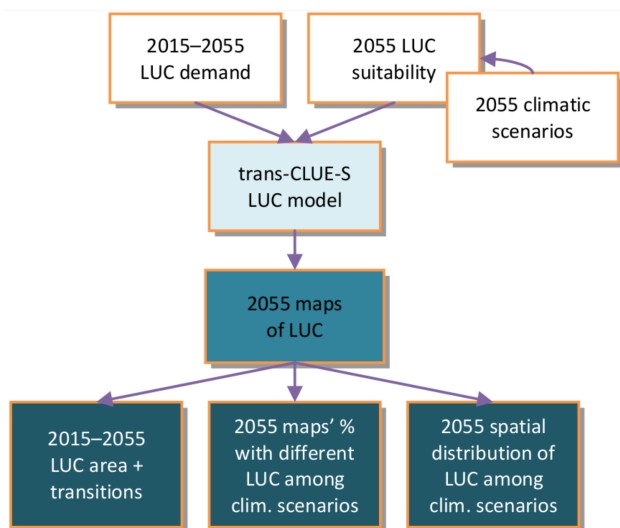

**Figure 2.** Basic workflow of our approach to predict and analyse future LUC in the study area. The same workflow was followed during the validation of the LUC change model against the reference map of the latest historical year 2015. From lighter to darker fill, the boxes represent different types of input or output: basic input for the LUC change model (white fill); LUC change modelling (light-shaded fill); maps predicted by the model (dark-shaded fill); and analyses of predicted maps (darker-shaded fill with bright letters).

A popular model of LUC change for the scale of our interest is the "Conversion of Land Use and its Effects at Small regional extent" (CLUE-S) [36]. The CLUE-S model is appropriate for finer spatial resolutions as of the present study's (map cells of ≲1 km), and for local to regional scales [37]. The model basically needs two types of input: (1) categorical maps of observed LUC at historical time points, with each cell covered by a single LUC type at each time point; and (2) biophysical and socioeconomic conditions for the respective cells and time points. The model's preparation consists of three steps [38]: calibration (via model parameterisation); prediction (via simulation); and validation (via quantitative assessment). It is recommended that the model is calibrated from all the time points except the last one, and then it is called to predict the map of the last time point, for finally validating the predicted map against the observed reference map of the last time point [38]. After an adequate number of calibration–prediction–validation attempts, the model is called to predict LUC at future time points under a specific climatic scenario and other biophysical and socioeconomic conditions. In the present study, we employed a more detailed variant of the CLUE-S model, i.e., the trans-CLUE-S model, which demonstrated higher predictive performance than CLUE-S [39]. The calibration of the trans-CLUE-S model, similar to its parental CLUE-S model, focuses on the two core components of the non-spatial demand of LUC transitions, and of the spatial allocation of that demand.

Regarding the non-spatial component of demand, trans-CLUE-S requires as input the number of cells in each LUC transition from a map at time step $t = 1$ (map 1) to the new map 2 to predict at a time step $t = 2$ [37]. A transition matrix can be built with cross-tabulation of the frequencies of LUC types between two consecutive maps of an earlier time interval, e.g., between the map 0 of a previous time step $t = 0$, and the map 1. An entry of the transition matrix holds the number of cells which transitioned from the LUC type in the row (map 0) to the LUC type in the column (map 1). If we divide each matrix entry by the sum of its row's entries (the total number of cells covered in map 0 by the row's LUC type), we get transition probabilities of a Markov matrix. For a LUC type in a row of the Markov matrix, we can estimate the demanded transitions in map 2 from map 1, by multiplying the row's probabilities by the known number of cells covered by the LUC type in map 1. In case the duration of the interval between map 1 and map 2 is different from the map 0–map 1 interval, it is possible to estimate the probabilities for the desired duration [40,41].

Regarding the spatial component of demand allocation, a statistical model is built for the suitability of each cell to each LUC type based on the environmental conditions at each cell of map 2. The statistical model is then used for allocating the demand of each LUC type transition to the most suitable cells in the map 2 under the presumed environmental conditions at $t = 2$.

### 2.3.1. Demand Scenarios

The three climatic scenarios were compared under the same demand scenario. Note that demand refers to the pre-specified demand in LUC and related LUC transitions in the future map to be predicted. We employed the following four demand scenarios:

1.  No demand. We compared the climatic scenarios under no demand restrictions. Without demand, the LUC transitions and total cover in the future could vary between climatic scenarios. Essentially, the no demand scenario returned a mere suitability map of the study area, since each cell was assigned to the LUC type with the highest suitability.

2.  Business-as-usual. We projected the 1996–2015 period's LUC transition matrix to the 2015–2055 period via a quadratic, regression-based estimation of the transition probabilities [40]. Before trusting the projection to the future, though, we did a validation test of the estimation method on the previous historical period of 1970–1996, to estimate and validate against the known 1996–2015 period (Figure S9). The absolute difference in the transitioned relative cover between reference and estimated map percentages was not greater than around 4% of the map, with the greatest differences being the overestimation of farmland persistence, as well as the underestimation of farmland becoming grassland and of forest persisting (Figure S10). This result was not surprising because in a previous work we found that farmland abandonment and subsequent succession accelerated from the 1970–1996 to the 1996–2015 period in our study area [20]. Thus, the projected 1996–2015 transitions to the future 2015–2055 were expected to carry the signs of this acceleration of farmland abandonment (Figure S11).

3.  As-usual, but with intensive farming preserved. To make milder the effect of accelerating farmland abandonment in the business-as-usual scenario, we kept the same demand scenario but preserved some of the 2015 farmland. The reason was that in our previous work in the study area, we found that the remaining farmland in 2015 was of intensive agriculture, being in the lowlands, in flatter ground, and with irrigation systems developed [20]. Thus, we selected from the 2015 LUC map the presumably intensive farming areas which could persist until 2055. We filtered this farmland on the basis of the elevation and slope distributions of the 2015 farmland (Figure S12). This was facilitated by the shape of the distributions, allowing us to keep any farmland which was on elevation no more than 420 m, and on slope not steeper than 10°. This 2055 farmland comprised 2.6% of the 2055 map, instead of the 0.4% of the business-as-usual scenario, and was located mainly on sites 3 and 5 (Figure S13). The relative cover of the other LUC types was predicted slightly less under this scenario for 2055 (Figure S14A).

4.  Extensive farming as in the 1970s. According to this optimistic scenario for demand, rural policies from 2015 onwards become very beneficial for the mountainous areas of the Mediterranean, supporting the extensification of agriculture, the return of the population and its occupation in local businesses, the increase of livestock, and the clearance of woodland and scrubland for once again becoming farmland and grassland [42]. Such characteristics of extensive agriculture were still prevalent in year 1970 in our study area [20,21]. Thus, we assumed that the relative cover of the land types in 2055 would be equal to their relative cover in 1970. For the land type transitions in the 2015–2055 period, we assumed that they would follow the reverse pathway from the 1970–2015. Thus, we only had to use the transposed transition matrix of the 1970–2015 time period as the 2015–2055 transition matrix of this demand scenario (Figure S14B).

2.3.2. Suitability Model

We related LUC suitability to biophysical and socioeconomic predictors with a Random Forest multiclass classification model. Our classification model returned the probability of each LUC type on each map cell given the predictor conditions in that cell. From the available predictor variables (Figure S1), we selected variables with lower inter-correlation (Spearman correlation coefficient $\leq 0.5$), but also higher potential for interpretation of results (Figure S15). Specifically, we used the elevation, slope, northness, eastness, presence of silicate and flysch parent rock, annual mean temperature, temperature seasonality, annual precipitation, precipitation seasonality, population and livestock density, and distance to the nearest settlement.

We used a Random Forest model because it does not have many parameters to calibrate, it is not liable to criteria concerning the distribution of values of the variables, and it can fit upon non-linear relations, unlike a linear model [43]. Additionally, in a spatial context as ours, a Random Forest does not require semivariograms which are difficult to model with their related assumptions [44]. Nevertheless, it is not advisable to apply plain machine learning models, such as a Random Forest, to spatio-temporal data similar to ours (map data from different years), because plain models ignore the common spatio-temporal autocorrelations. Without paremeterising such models properly, there is a high chance of overfitting, and overestimating the model's predictive performance [45]. To avoid such issues, we tested the parameterisation of the Random Forest model via different cross-validation methods with the R package "CAST" [46]. Specifically, we initially compared the performance of random, leave-location-out, leave-time-out and leave-location-time-out cross-validation schemes on the data from 1945, 1970 and 1996 (Figure S16). Locations were grouped according to the five sites, and time according to the three years. The cross-validation performance of the random partitioning was higher than the rest of the schemes (Figure S16). Nevertheless, when called to predict data unknown to the training, from the 2015 map, the performance of random cross-validation was similar to the other cross-validation partitioning schemes (Figure S17), demonstrating that we would have overestimated model performance if we had not taken into consideration spatiotemporal autocorrelations. Thus, comparing the cross-validation performance of the full models under the four schemes, i.e., when data from also 2015 were used in the training, again random cross-validation had superior performance (Figure S18). Given the previous validation exercise (Figure S17), we nevertheless know now that performance when predicting new map data, such as for the suitability of 2055 map, would be similar to the non-random cross-validation schemes, i.e., values around 0.63 for the measure of the Area Under the ROC Curve (AUC). We chose as performance measure the Area under the Receiver Operating Characteristic (ROC), which is the curve that relates the True Positive Rate and False Positive Rate of a binary classifier across different discrimination thresholds, because it takes into account the trade-off between sensitivity and specificity at the best-chosen threshold, and it is comparable between different models and scenarios. A larger AUC to a maximum of 1 denotes better performance, with the minimum of 0.5 denoting performance not better than a randomly guessing model.

Thus, the Random Forest model was parameterised and fitted on a balanced subset of the whole dataset, i.e., a training dataset of $n$ randomly selected observations. Specifically, for any combination of LUC type, site, and year, the minimum number of observations was 92. Hence, we randomly selected 92 observations from each combination, leading to a training set with $n = 92 \times 5 \times 5 \times 4 = 9200$ observations. With the R package "caret" [47], hyperparameters were fine-tuned by the leave-location-time-out cross-validation scheme. We tried the different combinations of split rule ("gini" or "extratrees") and number of randomly selected predictors at each split (two; half or all of the variables), choosing the combination which maximised performance in terms of the AUC. The hyperparameter for the minimum number of observations in the terminal nodes of individual trees was fixed and equal to one. Finally, the trees count was fixed at 1000, since it is not necessary to fine-tune it [48].

A relationship between LUC type occurrence and a predictor was investigated with the predictor's average of individual marginal effects [49]. An individual effect for a predictor's value was the Random Forest-predicted value of the response when the other predictors could take one of their $n$ value combinations in the training data. Thus, for the range of a predictor's values, a partial dependence plot presented the LOESS of the average among the $n$ individual effect curves. We furthermore inspected a sample of the $n$ curves, to confirm the plausibility of the average curve (Figure S19). Only the predictors with the most representative average curves were chosen for display. We produced the data for the plots with the R package "pdp" [50].

### 2.3.3. Allocation of Demand

As said previously, we used a variant of the CLUE-S model [36], the trans-CLUE-S [39]. Their main difference is that trans-CLUE-S requires demand at the level of LUC type transitions from the previous to the next map (all entries from a transition matrix), whereas its parental CLUE-S requires demand at the level of LUC type total cover in the next map (only the column sums of the transition matrix). Additionally, both models require the suitability matrix as input, where each matrix entry contains the probability (suitability) of the cell at the row for the LUC type at the column. The basic CLUE-S allocation algorithm assigns to each cell the LUC type with the highest suitability. If the number of cells assigned to a LUC type deviates from the LUC type's demanded total cover, then the algorithm iteratively alters the suitability of all cells for that type, scaling this alteration proportionally to the deviation. The iterative alteration of suitability stops when demand is satisfied to a desired deviation distance for all LUC types. The trans-CLUE-S essentially runs a CLUE-S allocation, but within each LUC type separately, instead of the entire landscape. That is, the simulation concentrates on the cells which were of a focal LUC type in the previous map. The demand for a focal LUC type in the next map is taken from the LUC type's row in the transition matrix. It then knows how many cells will persist (the row's entry at the main diagonal), and how many cells will turn to other LUC types (off-diagonal entries). As in CLUE-S, it alters iteratively the suitability of the deviated LUC types until demand is satisfied to a desired deviation for the focal LUC type. The same routine is executed for the cells of the other LUC types in the previous map. Another work is dedicated to the description and testing of the trans-CLUE-S model [39].

To fully meet the specified demand, we appended to the trans-CLUE-S allocation routine the sub-routine of the college admissions problem, which has been shown to facilitate and speed up the convergence of the allocation [39]. Additionally, we did not incorporate any CLUE-S model constraints in the five transition rules, or in the elasticity settings. Regarding the five transition rules, we did not prevent any cell's LUC type from changing: (1) completely throughout space and time; (2) if it has not persisted for a minimum number of time steps; (3) if it has changed for a maximum number of steps; (4) outside the LUC type's defined spatial neighbourhood; and (5) in specific localities throughout time. For elasticity, no LUC type was more elastic to change than others.

We validated the trans-CLUE-S model's predictions against the observed map of 2015, after calibrating it with data from years 1945, 1970 and 1996, using as demand the following four cases: (1) the actual, observed transition matrix of the 1996–2015 period; (2) no demand; (3) business-as-usual from the 1970–1996 to the 1996–2015 period; and (4) business-as-usual but with intensive farming persisting. The first case was expected to deliver the best predictions, since the actual demand was used, without any demand estimation error involved. The model's performance in the following three cases, then, would be lower in relation to error originating mainly from the demand component. The three last cases had equivalent settings as described about the demand scenarios for predicting LUC in 2055, but they were calibrated without the use of data from year 2015. The same applies also for the suitability model, which was parameterized with data from years 1945, 1970 and 1996. The validation procedure showed that the predictions under the estimated demand scenarios had a decrease of around 7% in performance, in comparison to the predictions

with the actual demand (Figure S20). Specifically, predicting the year 2015 with the observed demand resulted in an AUC = 0.73 and to identical LUC to the observed 2015 map in 73% of the predicted map, whereas under the estimated demand scenarios the AUC = 0.7 and the match was 63–65% of the predicted map. Under no demand, the mere suitability map predicted for 2015 had an AUC = 0.62 and a 35% match. This worst performance, which was found under the no-demand scenario, demonstrated the usefulness of the trans-CLUE-S model's allocation of demand for delivering LUC predictions of 80% increased accuracy from a mere suitability map which omits any demand information.

*2.4. Comparisons between Climatic Scenarios*

We compared the predictions between two climatic scenarios under the same demand scenario in two ways. First, by calculating the percentage of the cells in the two compared maps which had different LUC allocated. Second, by comparing the biophysical and socioeconomic characteristics of LUC occurrence at these different parts of the two compared maps.

For the difference in allocation between climatic scenarios, we merely calculated the proportion of cells with different LUC. As a reference to this comparison, we additionally calculated the theoretical least and greatest differences that would be possible for the given demand scenario of LUC transitions. We adopted a simple approach that quantifies these lower and upper bounds of differences due to spatial allocation with $J$ LUC types, given all possible transitions of LUC type $i$ to $j$ between two demand scenarios, $A_{ij}$ and $B_{ij}$ [51]. Since we only compared climatic scenarios under the same demand scenario $A$, the least possible difference $L$ is 100% minus the greatest possible agreement which was 100%: $L = 100 - \sum_{i=1}^{J} \sum_{j=1}^{J} \min(A_{ij}, A_{ij}) = 100 - \sum_{i=1}^{J} \sum_{j=1}^{J} A_{ij} = 100 - 100 = 0$. The greatest possible difference $G$ was 100% minus the least possible agreement: $G = 100 - \sum_{i=1}^{J} \sum_{j=1}^{J} \max(0, 2A_{ij} - \sum_{j=1}^{J} A_{ij})$.

For the difference in LUC-type occurrence between climatic scenarios, we focused on the parts of the two compared maps that differed in LUC between the two extreme scenarios SSP-SUST and SSP-FUEL. For each LUC type, we then compared the distributions of suitability predictors in those cells covered by that LUC type in the two compared maps. Specifically, we compared the distributions of the predictors which were fixed in time and were found to be important in the classification of a cell's LUC type: elevation, slope, northness, population density, livestock density and settlement proximity. Due to outliers in the distributions, we statistically compared the median of the two distributions of a predictor from the two different climatic scenarios with the Wilcoxon signed-rank test. Additionally, we ran a Multiple Factor Analysis (MFA), to explore possible relationships between all LUC types, predictors and demand scenarios with the R package "FactoMineR" [52]. In the MFA, LUC types were assumed to be the individuals, and the same predictor from the different demand scenarios formed a set of variables. The quantitative value characterising an individual LUC type of a demand scenario was the Wilcoxon test's estimated difference in the median of a predictor's distribution under the climatic scenario SSP-FUEL minus its median under the SSP-SUST. Hence, a positive difference implied that the cells covered by that LUC type had higher values in the least optimistic scenario under a specific demand scenario. We enquired the biplot showing on the first two dimensions of the MFA both the position of the LUC types and the predictor centroids among the demand scenarios (the MFA produced four vectors per predictor, one for each demand scenario). The predictor vectors were rescaled for plotting purposes, i.e., they had a different unit of measurement than the LUC types on the MFA space; hence, only their direction mattered for the interpretation of the MFA. Specifically, the more a LUC type was positioned to the direction of a predictor, the higher the predictor's median was under the least optimistic SSP-FUEL in comparison to SSP-SUST.

## 3. Results

### 3.1. Demand Scenarios

The scenario of no demand for the 2015–2055 LUC transitions was the only scenario in which the 2055 relative cover of the LUC types could freely vary between climatic scenarios because there were no requirements for specific relative cover or land type transitions in the 2015–2055 period. Under no demand, thus, relative cover varied but less than 2% of the map for any LUC type between climatic scenarios (Figure 3A–C). Specifically, the greatest difference between climatic scenarios was for farmland, which was predicted suitable in 10.8% of the map under the most optimistic scenario SSP-SUST (Figure 3A), while farmland suitability was 12.7% and 12.5% under the less optimistic scenarios SSP-RIVAL and SSP-FUEL (Figure 3B,C). In general, the predicted LUC type relative covers of SSP-RIVAL and SSP-FUEL were more similar than between any of them and the SSP-SUST. The major transition towards the 2055 relative cover was of forest becoming scrubland (15.9% on average among climatic scenarios), grassland (11.6%) and farmland (3.1%), in order of decreasing percentage of the map, leaving forest in the 31% of the map (Figure. 3A–C). A similar grassland cover of 3.5% transitioned to farmland as well.

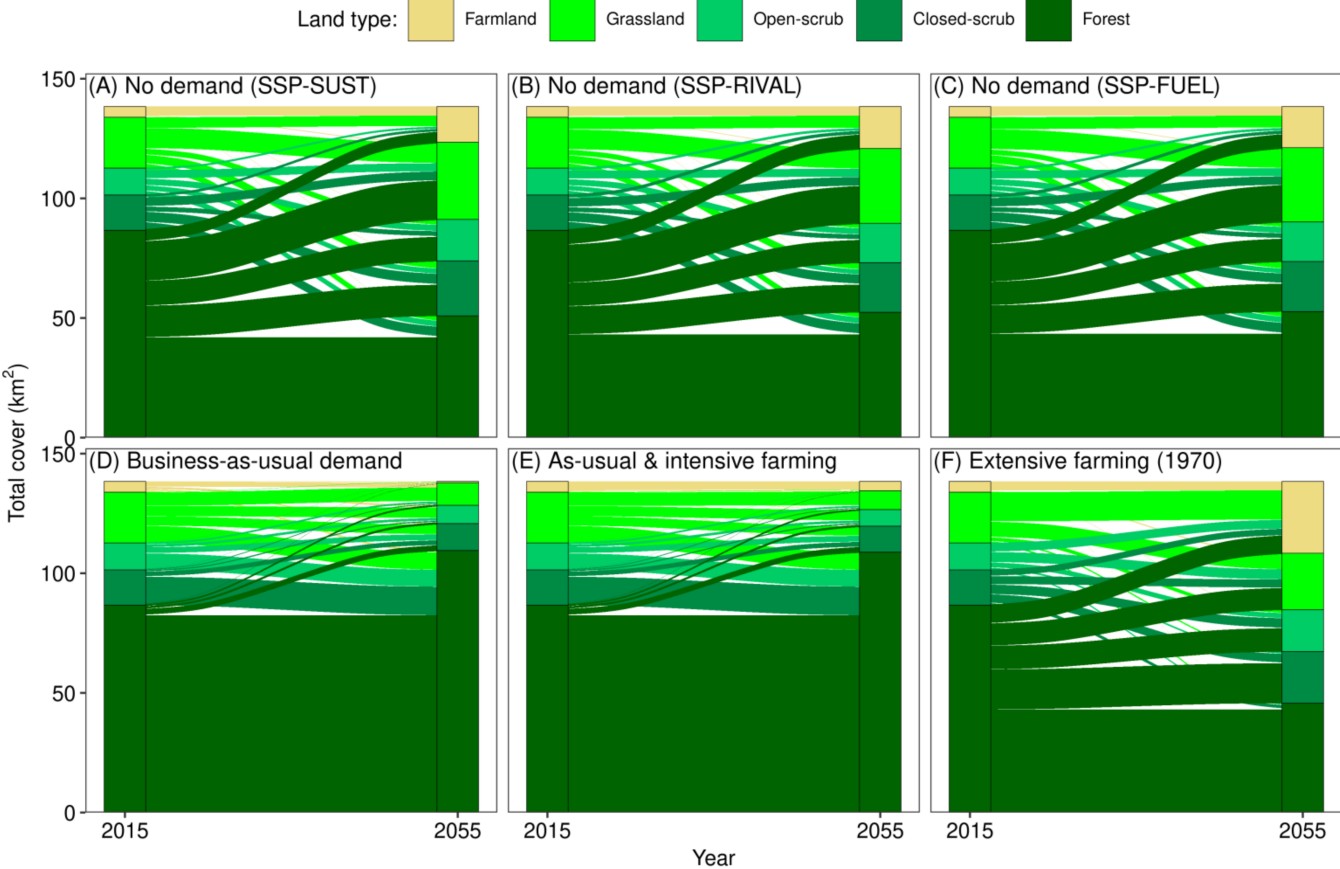

**Figure 3.** The LUC type cover and transitions over the 2015–2055 prediction period. The bars show the cover of each LUC type. The flows show the LUC transitions during the: (**A**) most optimistic climatic scenario with no pre-specified demand in 2015–2055 LUC transitions; (**B**) intermediate climatic scenario with no demand; (**C**) least-optimistic climatic scenario with no demand; (**D**) business-as-usual pre-specified demand from 1996–2015 to 2015–2055; (**E**) business-as-usual pre-specified demand but with the 2015 intensive farming preserved; and (**F**) pre-specified inverse transitioning to 1970 which had characteristics of extensive farming.

In the business-as-usual scenario for demand, the farmland's dramatic decrease from the 1996–2015 interval continued until 2055, resulting in a farmland relative cover of 0.4% in 2055 from 3.2% in 2015 (Figure 3D). Forest kept increasing as the LUC type with the largest share, estimated to cover 79% of the map by 2055, covering 62.6% in 2015. Farmland lost most of its cover to grassland (1.4%), whereas forest gained mainly from closed-scrub (8.6%), grassland (5.3%) and open-scrub (5%), in order of decreasing percentage of the map. Most of the LUC types transitioned to the direction of progressive succession, and the largest transition towards retrogressive succession was forest becoming closed-scrub in 1.4% of the map.

The other two demand scenarios were respectively similar to the two described previously. On the one hand, the scenario of preserving the intensive farmland of 2015 was the same as the business-as-usual, with the exception that 2.6% of the map persisted as farmland, instead of the only 0.2% persisting under business-as-usual (Figure 3E). On the other hand, the scenario of inverse transition to the 1970s relative cover of LUC types was similar to the no-demand scenario in which LUC types were allocated only according to environmental suitability. In specific, no LUC type transition differed more than 5% of the map between these two demand scenarios, with half of them less than 1% (Figure 3F). Thus, the relative cover of the LUC types in 1970 was similar to what was predicted for 2055 only on the basis of environmental suitability, with the exception of the greater cover of farmland at the expense of forest.

### 3.2. Allocation of Demand

Slope was the most important environmental factor of the Random Forest model for classifying a cell's LUC type based on its environmental conditions, and hence, for estimating LUC type suitability, which was used for demand allocation by the trans-CLUE-S model (Figure 4). Aside from variable importance, slope also had the greatest effect on LUC type suitability, i.e., leading to the greatest variation in the predicted probabilities of LUC type occurrence. In specific, suitability for farmland was decreasing for slopes up to 30°, whereas this effect was weaker and even positive for LUC types further in the sequence of vegetation succession (Figure 4A). A similar but smaller effect on farmland suitability had the elevation up to 750 m, with the rest of LUC types increasing their occurrence slightly (Figure 4B). Livestock had the opposite effect, increasing the suitability for farmland in densities of up to 300 animals $km^{-2}$, with this effect being weaker and even negative for more progressed vegetation types (Figure 4C). Other environmental factors with considerable effect were settlement proximity, northness, and human density, which all affected mostly forest and grassland in opposite directions for each variable (Figure 4D,F,J). Finally, all four bioclimatic variables were included in the top 10 most important variables, with annual mean temperature related positively with farmland and forest but negatively with the other LUC types (Figure 4E), and with annual precipitation and its seasonality, respectively related negatively and positively with farmland occurrence mainly (Figure 4G,H).

### 3.3. Difference in Allocation between Climatic Scenarios

With the trans-CLUE-S model's accuracy estimated to 63–65% in our validation tests (Figure S20), the pairs of maps predicted under two different climatic scenarios differed in less than 10% of their total area for any demand scenario (Figure 5). For any demand scenario, differences in allocation were greater for the comparisons between SSP-SUST versus SSP-RIVAL or SSP-FUEL, than for the SSP-RIVAL versus SSP-FUEL comparison. Between demand scenarios, differences were smaller for the demand scenarios of business-as-usual and of as-usual with persistent intensive farming since 2015 (Figure 5B,D). Taking into account the theoretical upper boundaries of the differences though (dotted lines in Figure 5), the differences in the predicted maps were similar across demand scenarios: close to 11% of the upper boundary in the comparisons between SSP-SUST versus SSP-RIVAL or SSP-FUEL; and close to 4% of the boundary for the SSP-RIVAL versus SSP-FUEL

comparisons. The upper boundaries were lower for the two demand scenarios of business-as-usual and of as-usual with intensive farming because they had greater LUC transitions and persistence in the 2015–2055 interval (Figure 3D,E) in comparison to the other scenarios (Figure 3A–C,F). Thus, these two demand scenarios limited the possibility for a wider range of spatial configurations.

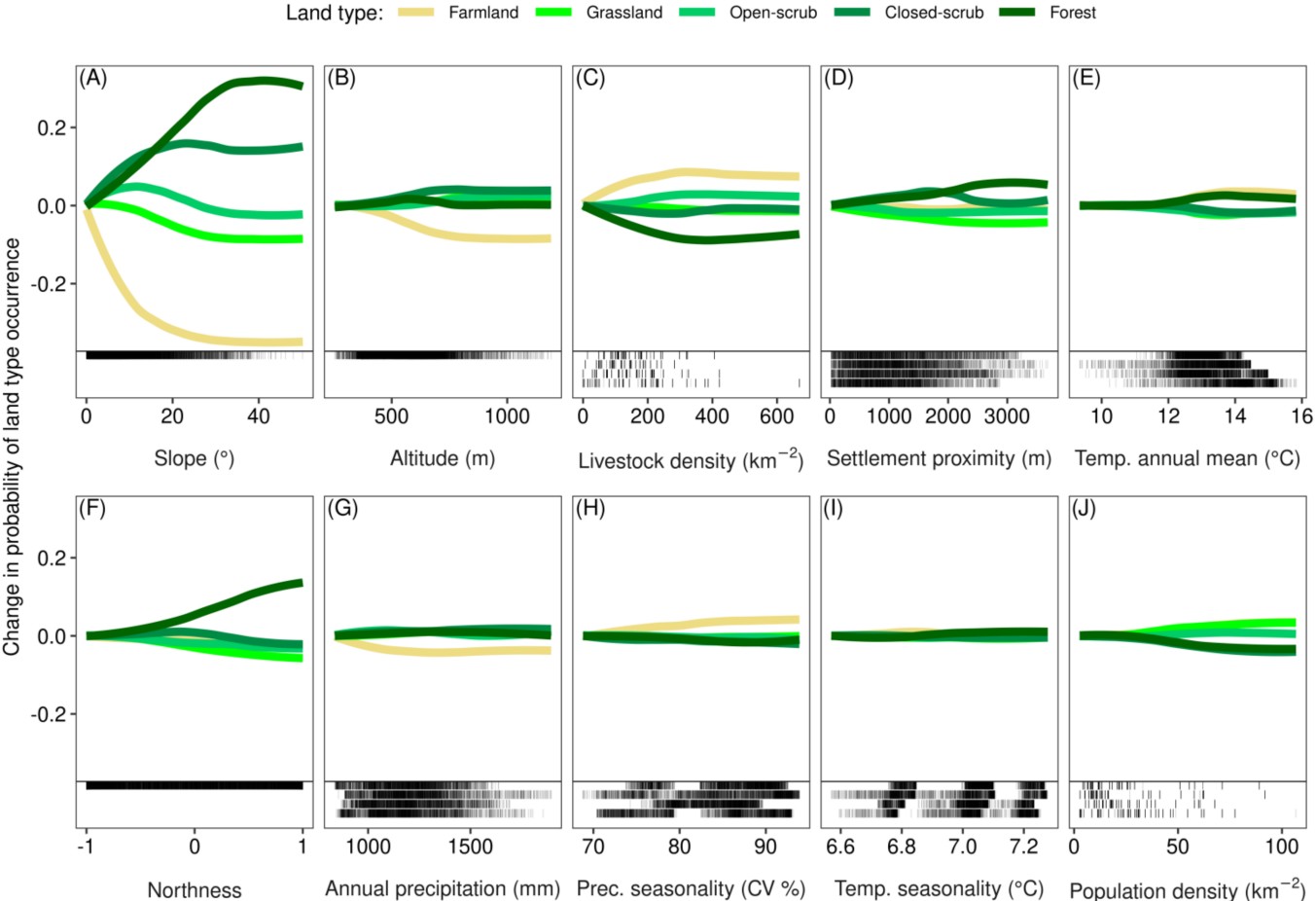

**Figure 4.** Mean marginal effects of the top 10 predictors ordered in decreasing importance for LUC type classification by the Random Forest model (panels (**A**–**J**)). For comparability, the curves are centred to the mean prediction for the left-most, minimum value of each predictor. The vertical marks under the horizontal line show the predictor's values in the training. If a predictor took different values in different years, the vertical marks are stacked in four rows, each row for a year, 1945 to 2015 from top to bottom.

Such small differences in the allocation of demand could be roughly identified visually on the maps from the different climatic scenarios (e.g., see Figure 6 for the business-as-usual demand scenario). Since slope was the most important factor for allocating demand to any predicted map by the trans-CLUE-S model (Figure 4), we can focus on map parts with steeper slopes (Figure 1). For example, the most northern part of site 1, which was covered by large and contiguous patches of closed-scrub and open-scrub in 2015 (Figure 6A), was predicted to be covered mainly by forest and small remnants of open-scrub and closed-scrub in 2055 by any climatic scenario (Figure 6B–D). Nevertheless, a closer look reveals that the less optimistic a climatic scenario was, the more the area that the forest covered at the expense of open-scrub and closed-scrub. Similar encroachment of closed-scrub by forest under less optimistic climatic scenarios can be seen in the steeper slopes of the mountain side west of site 5 (Figure 6B–D).

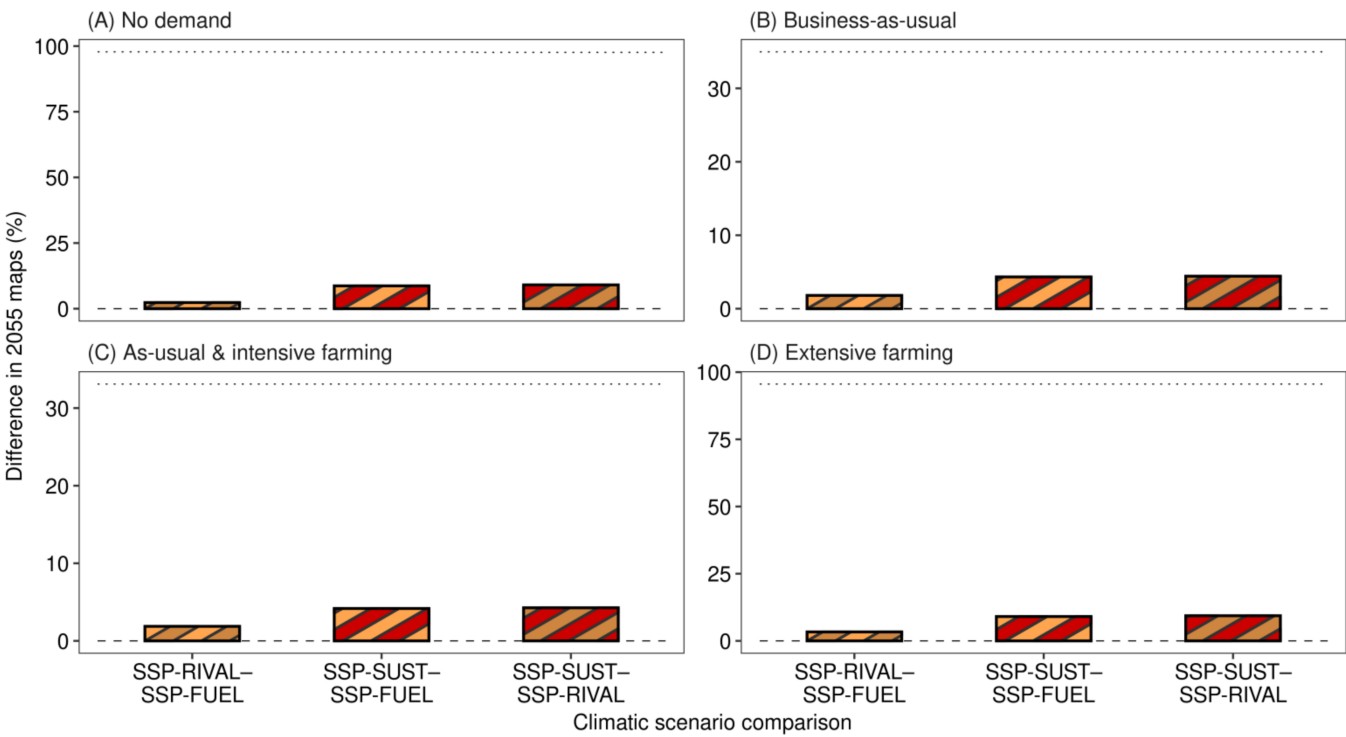

**Figure 5.** Percent of the map cells that had different LUC in two predicted maps of 2055 under two climatic scenarios, and for any of the following four scenarios of demand in 2015–2055 LUC transitions: (**A**) no pre-specified demand; (**B**) business-as-usual pre-specified demand; (**C**) as-usual, but with the 2015 intensive farming preserved; and (**D**) inverse transitioning to 1970, which had characteristics of extensive farming. The three climatic scenarios were the SSP-SUST, SSP-RIVAL and SSP-FUEL, from the more to the less optimistic one. The horizontal dashed and dotted lines indicate the respective lower and upper bounds which are the theoretically least and greatest differences that the two compared maps can have for the occurred 2015–2055 LUC transitions of the specific demand scenario.

*3.4. Difference in LUC Type Occurrence between Climatic Scenarios*

Since the largest of the otherwise small differences between maps predicted under different climatic scenarios were between the most optimistic SSP-SUST and the less optimistic SSP-RIVAL or SSP-FUEL (Figure 5), we limited our further investigation to comparisons between the extreme climatic scenarios SSP-SUST and SSP-FUEL. Regarding slope again, which was the most important factor for allocating demand to any predicted map by the trans-CLUE-S model (Figure 4), we focused on its values at the map cells which had different LUC between the maps predicted by SSP-SUST and SSP-FUEL. Confirming the previous visual observations (Figure 6), the slope in such map cells covered by forest and closed-scrub exhibited consistent trends across all demand scenarios (Figure 7). In specific, forest under SSP-FUEL was predicted in cells with significantly steeper slope than forest under SSP-SUST, whereas closed-scrub was predicted in cells with significantly less steep slope ($p \leq 10^{-3}$; Wilcoxon test). The rest of LUC types exhibited less consistent trends in the slope of the map cells they were predicted to cover. We carried out similar tests for other environmental predictors (Figures S21–S25).

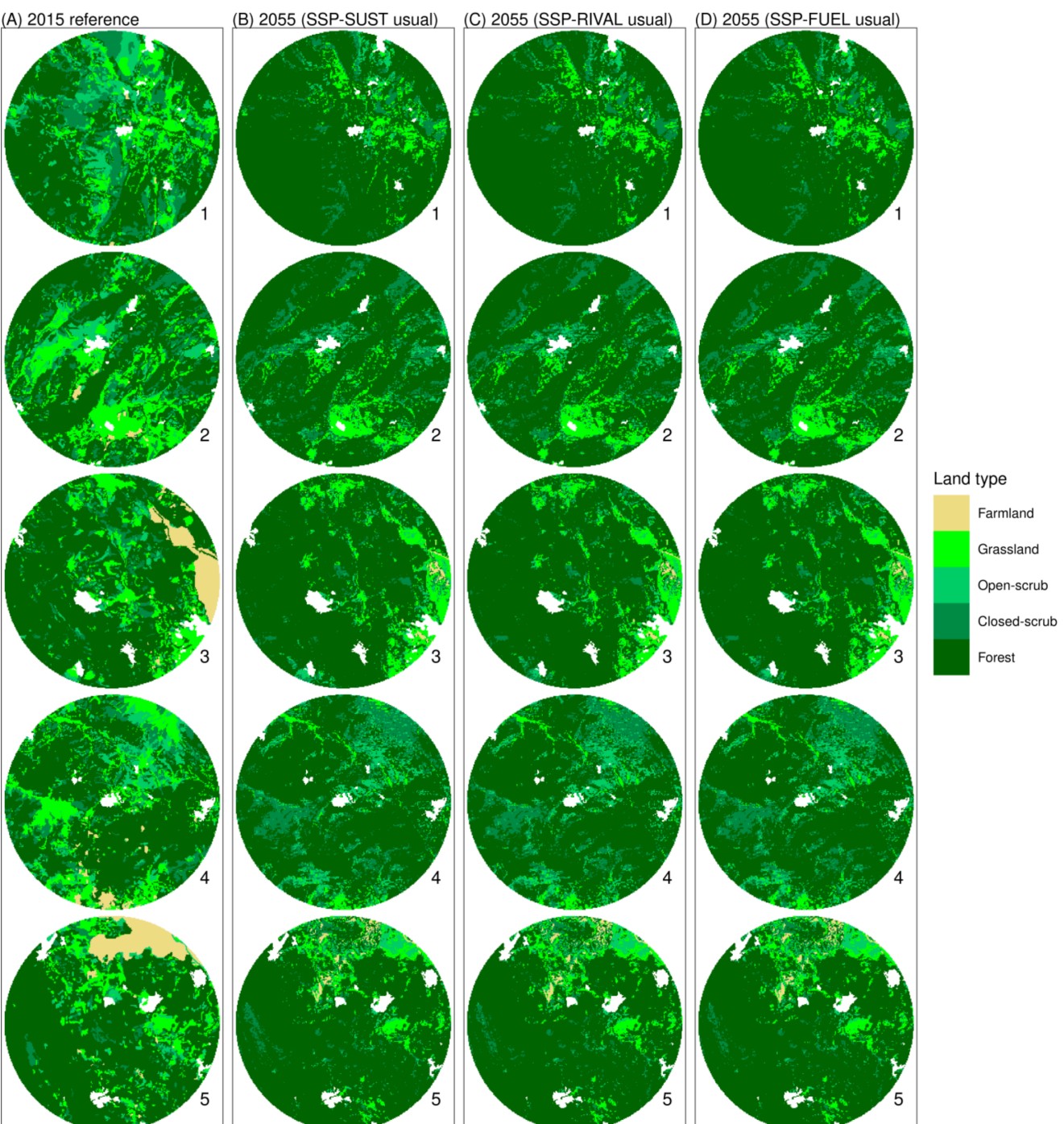

**Figure 6.** Future LUC predictions for the three climatic scenarios under the business-as-usual demand scenario. For reference, we provide the map of the year 2015 (**A**). The three climatic scenarios were the SSP-SUST (**B**), SSP-RIVAL (**C**) and SSP-FUEL (**D**), from the most to the least optimistic one. The five study sites of 6 km diameter are numbered as in Figure 1, maintaining the same orientation vertically towards north.

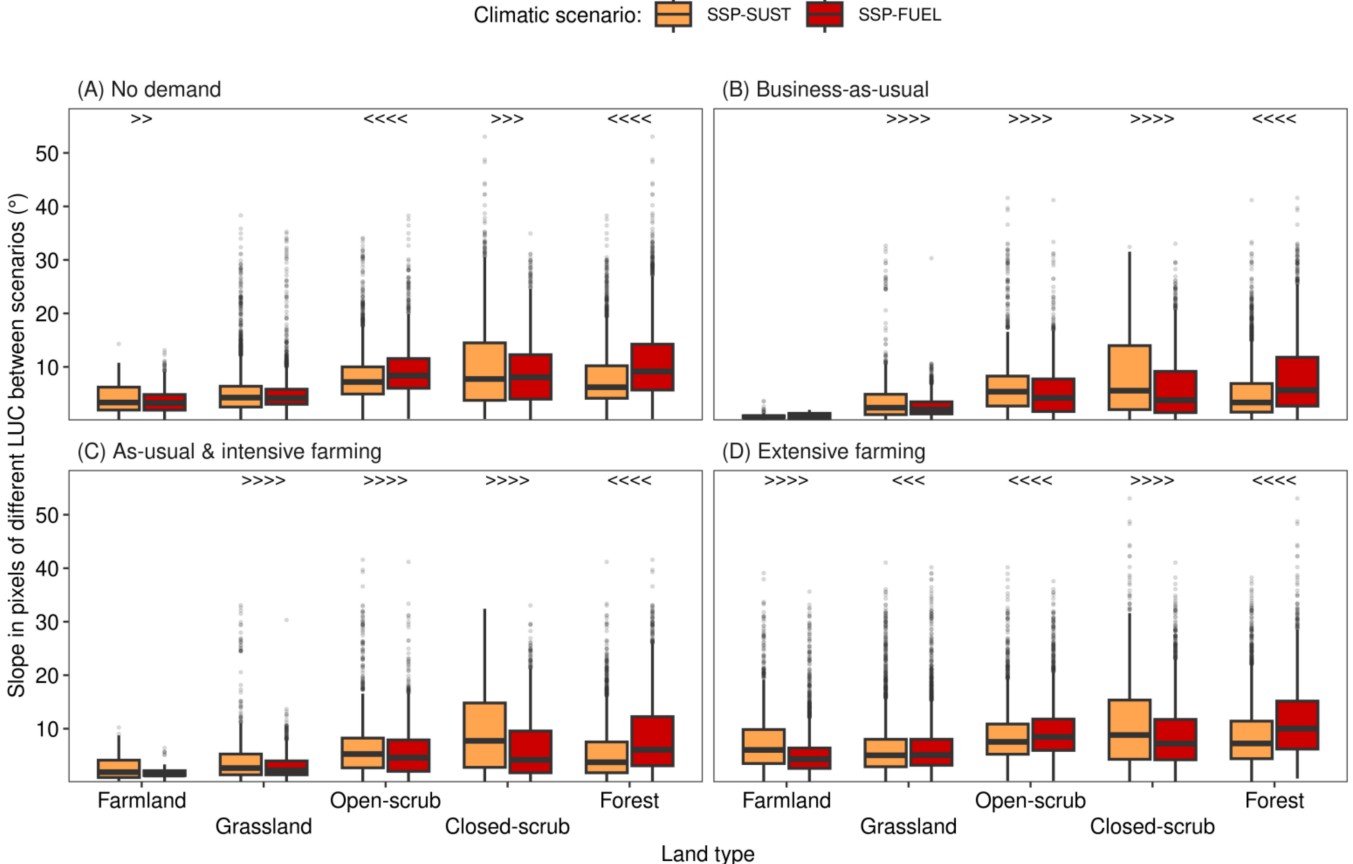

**Figure 7.** Slope of LUC type at the map cells which differed in the prediction of LUC between the most optimistic (SSP-SUST) and most pessimistic (SSP-FUEL) climatic scenario, under each of the following four scenarios of demand in 2015–2055 LUC transitions: (**A**) no pre-specified demand; (**B**) as-usual from 1996–2015 to 2015–2055; (**C**) as-usual, but with the 2015 intensive farming preserved; and (**D**) inverse transitioning to 1970 which had characteristics of extensive farming. Inequality symbols above the boxplots indicate statistically significant difference in the median slope by Wilcoxon test between scenarios. In specific, four inequality symbols were used for $p$-value $\leq 10^{-4}$, three symbols for $p \leq 10^{-3}$, two for $p \leq 0.01$, one for $p \leq 0.05$, and no symbol for $p > 0.05$ level of statistical significance.

We finally summarised similar trends under all four demand scenarios with the MFA for not only slope, but for all six environmental predictors which were fixed in time, and which were the most important for the suitabillity allocation of demand (Figure 4). The first two dimensions of the MFA retained 74.3% of the total variance (Figure 8). A closer distance between a LUC type and an environmental predictor on the factor map meant that the LUC type was predicted at map cells with higher values of the environmental factor under the climatically less optimistic SSP-FUEL scenario. The first MFA axis was more strongly related to the remoteness of the map cells from settlements, and with population density. In specific, farmland was predicted in more remote areas under SSP-FUEL, whereas grassland and open-scrub were predicted closer to settlements but in municipal districts with lower population density under this least optimistic climatic scenario than under the most optimistic one. The second MFA dimension was more related to slope, elevation, northness and livestock density. In specific, forest was similarly to slope predicted in higher elevation and northness, whereas closed-scrub was predicted in lower values of these three predictors, but in municipal districts with higher livestock density (Figure 8).

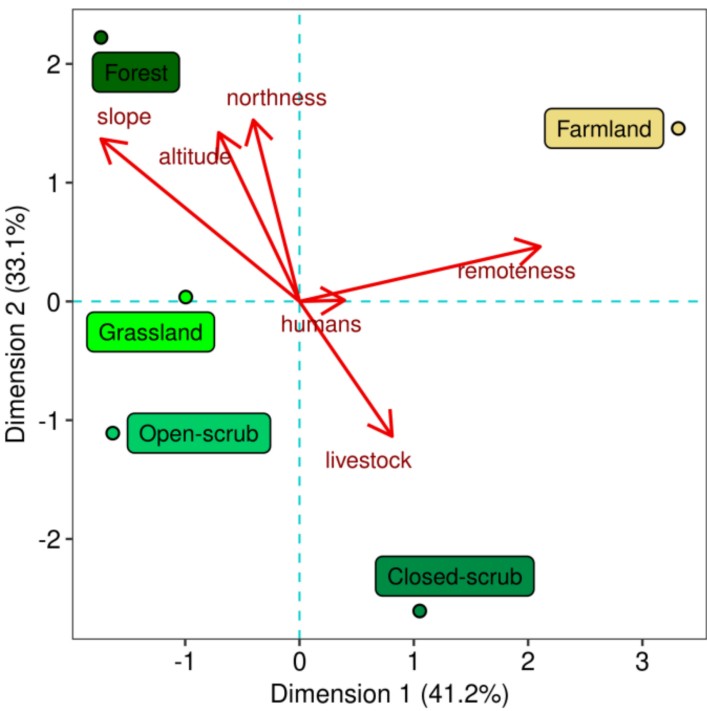

**Figure 8.** The change in LUC type conditions when moving from the most (SSP-SUST) to the least (SSP-FUEL) optimistic climatic scenario. In this factor map from the MFA, a more similar direction between a LUC type and an environmental predictor means that the LUC type was predicted at map cells with higher values of the environmental factor under the least optimistic climatic scenario. The MFA regarded only the map cells which differed in the prediction of LUC between the most and least optimistic climatic scenarios. We employed the six environmental factors which were fixed in time and the most important in the spatial allocation of demand. The MFA took into account all four scenarios of demand in the 2015–2055 LUC transitions, grouping the same environmental factor across the demand scenarios (i.e., six sets of four variables). The factor map's axes provide the percent of variance retained by the first two dimensions of the MFA.

## 4. Discussion

The aim of the present study was the investigation of climatic effects on the future LUC of a typical mountainous area of the Mediterranean. We found that climate change played a minor role in the predictions of the LUC demand, and of its spatial allocation to the 2055 maps. In specific, under no limitation of demand in LUC transitions, no LUC type transition differed by more than 2% of the map between climatic scenarios. Under the business-as-usual demand scenario, farmland nearly disappeared, and forest further expanded. Additionally, pairs of maps predicted under two different climatic scenarios differed in less than 10% of the maps for any demand scenario. Nevertheless, we found statistically significant environmental differences in the spatial allocation of the LUC in the less than 10% that differed, such as the spatial shift of farmland and forest to higher elevation under such a warmer and drier climate. We discuss our findings in the context of anticipated changes in climate and LUC at the Mediterranean mountains during the 21st century.

To address the fullest possible range of climatic scenarios, we employed Shared So­cioeconomic Pathways from sustainability (SSP-SUST) to regional rivalry (SSP-RIVAL) to fossil-fuelled development (SSP-FUEL) [29]. We employed the bioclimatic variables of annual mean temperature, temperature seasonality, annual precipitation and precipitation seasonality (Figure 4). These variables have been investigated in climate projections for mountainous regions and for the 2055 horizon we studied, especially the annual mean temperature and annual precipitation [8]. The values of the latter two variables that we

downscaled from the Chelsa dataset of 1 km resolution to our 25 m raster data are close to the ones reported for Mediterranean mountains in 2055 under the extreme scenarios [8]. In specific, the latter authors found that under their most and least optimistic scenarios, the expected warming rates were, respectively +2.3 °C and +3.2 °C for the 1970s–2050s period, in comparison to our rates of +1.8 °C and +3 °C under SSP-SUST and SSP-FUEL, respectively. For this period, their estimated change in the annual precipitation in Mediterranean mountains under the most and least optimistic scenarios was, respectively −2% and −5.9% [8], in comparison to the −3.1% and −8.9% from our dataset. These changes in temperature and precipitation, even under the most optimistic scenario, are commonly larger than the ones reported in lowland regions of mid-latitudes around the globe, especially regarding temperature [53]. We hence expected that climate could have a strong effect on the predictions of LUC in our study area.

Nevertheless, pairs of climatic scenarios resulted in a less than 10% of predicted maps having different LUC, which was around 11% of the greatest possible differences given their common demand scenario (Figure 5). Similarly, previous simulation models showed that climate has a smaller effect than LUC on the prediction of the broad habitat types employed therein, and which are similar to our study's LUC types [11]. In terms of indicators (of LUC, biodiversity, and ecosystem services), comparisons between projection models have also shown similar values between different climatic scenarios but under the same socioeconomic conditions, similar to our study's setting of comparing climatic scenarios under the same demand scenario [54]. Besides future projections, climate has been shown to historically have a smaller effect than LUC on the forest encroachment process of forest–pasture ecotones at the treeline of Mediterranean mountains [19]. The technical reason for the smaller climatic effect in our study appeared to be the lower importance that the four bioclimatic variables had in the suitability of the cells to the different LUC types by the Random Forest model (Figure 4). Lower importance resulted in smaller changes in LUC suitability under different climatic scenarios, since only the values of the four bioclimatic variables changed between two compared climatic scenarios. In particular, farmland occurrence was the LUC type with the greatest influence with these variables. This result can be related to the larger contribution of farmland to the first dimension of the MFA (Figure 8). Additionally, it can explain why farmland relative cover exhibited the greatest differences between climatic scenarios under the no demand scenario (Figure 3). This scenario was important for our comparisons, because it was the only scenario in which LUC demand was free to vary between climatic scenarios, to uncover any unconstrained effect of climate on LUC predictions.

Besides the no demand scenario, the other demand scenarios facilitated the more controlled investigation of climate's effect on LUC. The specification of demand before the simulation of future LUC is a characteristic of many LUC models like the one employed herein [37]. Other models are cross-sectoral, without the beforehand specification of demand, but with the integration of interactions between socioeconomic factors which are climate-related, such as wood and irrigation demand [54,55]. Such models have been shown to return safer predictions of LUC than simpler models like trans-CLUE-S. Nevertheless, demand for the future LUC in models similar to trans-CLUE-S is estimated by the projection of the transition matrix from a previous time interval, of relevant time series, or of economic models [37,56,57]. This feature of pre-specification of demand in such spatial models can be convenient for studies of small spatial scales as of our study. One reason is that it reduces the model's complexity, the data requirements especially from data-poor regions, and the computational resources. Furthermore, effects of socioeconomic sectors acting in a broader scale which can impact LUC at the local scale are implicitly included via the recent trend projection that leads to the estimation of demand. In that way, climate-related socioeconomic sectors such as wood and irrigation demand are implicitly taken into account via the demand scenario, instead of explicitly formulating them as part of the model [37,58].

Such an example is our demand scenario of business-as-usual which predicted further farmland abandonment and forest expansion before simulating the spatial allocation of this change (Figure 3D). Similar abandonment trends have been predicted by models integrating different socioeconomic sectors explicitly into their allocation simulations, without pre-specified demand, relating abandonment to decreases in the productivity and profitability of the agricultural sector due to climate-related heat and drought increases [54,55]. Interestingly, predictions of abandonment and extensification in southern Europe have been robust among different types of models, and different factors incorporated in the models [15,54,55,59]. In our study, the 2015–2055 transition demands, such as in the business-as-usual scenario, were projections of the 1996–2015 period, during which an acceleration of farmland abandonment was identified, together with a subsequent increase in the rates of secondary succession [20]. This acceleration of abandonment was related to a significantly higher decrease of the population and livestock densities in the municipalities during that period, in comparison to previous periods. Socioeconomically, the declines in population and livestock can be interpreted as the echo of the mistargeted policies for low-intensity farming during the 1970–1996 previous period [21], and as a result of the financial crisis of the 1996–2015 period [60].

With a plausible demand scenario determined before the simulation, any effects of climate on the spatial allocation of demand were expected to be more easily highlighted. Although the maps between different climatic scenarios under the same demand scenario were identical in more than 90% of their area, there were significant differences in the occurrence of LUC types in the remaining 10% that approximately differed. As already mentioned, this lower effect of climate on LUC can be technically related to the lower importance that the bioclimatic variables had in the Random Forest sub-model of suitability (Figure 4). A lower relative importance meant that different values of the four bioclimatic variables, which were the only variables that changed between climatic scenarios, resulted in smaller suitability changes, and hence to fewer differences in the suitability-based, spatial allocation of the same demand. In less optimistic scenarios for climate, annual mean temperature and temperature seasonality overall increased in our data, whereas annual precipitation and precipitation seasonality decreased (Figure S2). According to the suitability model, an increase in temperature and its seasonality were related to increased suitability for farmland and forest, and a decreased suitability for grassland and scrubland (Figure 4E,I); the decreases in precipitation and its seasonality were related to a respective decrease and increase in mainly farmland suitability (Figure 4G,H). Given that among these four bioclimatic variables, mainly annual mean temperature correlated positively and annual precipitation correlated negatively with elevation (Figure S15), we can relate the climate-related changes in suitability with elevation-related positions on the predicted maps for convenience, to more clearly interpret the MFA biplot which constituted a summary of our results (Figure 8).

The first dimension of the MFA could be related more to farmland, grassland and open scrub from the LUC types, and mainly with remoteness from the suitability predictors to the direction of these LUC types. In specific, farmland was predicted to spatially shift away from settlements under the least optimistic scenario, whereas grassland and open-scrub came closer. Since farmland suitability could increase with the increase in temperature and the decrease in precipitation (Figure 4E,G), this could mainly occur in higher elevation where distances to nearest settlements are greater (Figure S15), because temperature and precipitation were, respectively, higher and lower in lower elevation already. The spatial shift could be inferred because, under the same demand scenario, each LUC type must occupy the same proportion in the area with different LUC between two maps predicted under different climatic scenarios. Thus, any differences in LUC must be due to spatial shift and swapping [61].

The second dimension of the MFA could be more related to forest and closed-scrub from the LUC types, and with the predictors of elevation, slope, northness and livestock density. Under a warmer and drier climate, forest was predicted to move at higher elevation,

slope, and to be more northern facing (Figure 8). Technically, these results are related to the higher probability of forest occurrence under these conditions, according to the suitability model (Figure 4A–C,F). An exception is elevation, for which forest occurrence was almost equiprobable across the elevation range, according to the suitability model (Figure 4B). Nevertheless, the trans-CLUE-S model was forced to spatially allocate more forest in higher elevation, showcasing that not only suitability, but also the competition between LUC types in the demand can determine spatial allocation [36]. The relocation of forest to sites with higher elevation and steeper slopes, as well as with more northern facing under a warmer and drier climate can be related to two aspects. On the one hand, forest retraction can be related to the vulnerability of lower elevation, milder slopes and less northern facing to moisture shortage, which has been shown to lead to decreased rates of secondary succession [62], and of tree growth [63], with closed-scrub taking the place, being more adapted to such conditions. On the other hand, the rising temperatures enable forest expansion in sites which were previously less suitable [64,65]. Nevertheless, it is advised that such forest transitions have to be interpreted in the context of LUC change in the Mediterranean mountains, since the anthropogenic abandonment of farmland and grassland has played a more important role in the shaping of the landscape [66,67]. The less than 10% difference in the maps between climatic scenarios demonstrated this concept in our study area.

## 5. Conclusions

The present study predicted that forest will further expand at the expense of the other LUC types in the landscape of a sub-mountainous area with characteristics similar to other Mediterranean areas. According to the demand scenarios estimated for the future, the primary role in shaping LUC was played by the abandonment of farmland and grassland. Secondarily, climate change was demonstrated to cause only minor shifts in the landscape, that is, with the shift of farmland and forest to higher elevation under a warmer and drier climate, and the shift of grassland and scrubland to lower elevation. These results indicate a strong potential for mitigation measures, given the influence of land abandonment and socioeconomic demands primarily, and climatic changes secondarily, with all three leading to rapid and relocating LUC change, both in the past [20], and in the future as shown herein.

The mitigation measures can be based in two insights provided by the present study. First, since land abandonment was found to be a stronger driver of LUC change than climate, mitigation measures would target in shaping LUC-related socioeconomic and political factors in favour of abandonment mitigation and even reversal, which can be more feasible than shaping climate-related factors. Second, moving from the broad types of LUC to the species level, the patterns, rates and predictors of LUC change can inform Species Distribution Models, for more accurate projections of future biodiversity in these ecologically and culturally characteristic but endangered landscapes of the Mediterranean.

Future studies can directly apply the present study's methodology for projections further to the future, i.e., to the 2071–2100 period for which climate projections are available in the CHELSAfuture dataset. It is expected that the uncertainty of the LUC projections will be higher in comparison to the 2055 horizon due to error propagation; hence, an appropriate method would be required for the estimation of the least and greatest differences possible [51]. Moreover, the LUC projections could be improved by additional socioeconomic data, e.g., from field sociological studies at the household level, in comparison to the municipality level of the present study. In a local context, different socioeconomic drivers could influence synergistically the projected LUC trajectories, which could be further aggregated in a broader context [68]. Conversely, socioeconomic drivers from a broader context, such as wood and irrigation demand, could increase the accuracy of spatial models, impacting LUC at the local scale of interest [54].

**Supplementary Materials:** The following supporting information can be downloaded at: https://www.mdpi.com/article/10.3390/land12010253/s1. Supplementary Materials: Supplementary Figures for the article (Figures S1–S25).

**Author Contributions:** Conceptualization, D.A.K., A.M., M.P., S.T., F.X., and I.T.; data curation, D.A.K., A.M., M.P., F.X., and I.T.; formal analysis, D.A.K.; investigation, D.A.K., A.M., S.T., F.X., and I.T.; methodology, D.A.K., A.M., M.P., F.X., and I.T.; software, D.A.K.; visualization, D.A.K.; writing—original draft, D.A.K.; writing—review and editing, D.A.K., A.M., M.P., S.T., F.X., and I.T.; supervision, F.X. and I.T.; funding acquisition, I.T.; project administration, I.T. All authors have read and agreed to this version of the manuscript.

**Funding:** The present study was supported by the Hellenic Foundation for Research and Innovation (H.F.R.I.) under the "1st Call for H.F.R.I. Research Projects to support Faculty Members & Researchers and the Procurement of High-cost Research Equipment Grant" (Project Number: 2333).

**Data Availability Statement:** The data which support the findings of this study are available from the authors upon reasonable request.

**Acknowledgments:** We acknowledge the Hellenic Cadastre, the Hellenic Military Geographical Service, and the Ministry of Rural Development and Food of the Hellenic Republic for providing orthophotos and large-scale aerial photographs. We would like to thank Georgia Bourdanou for the entry of socioeconomic data, Grigorios Vassilopoulos for organising land cover mapping, Anastasios Zotos for gathering socioeconomic data, and Ioannis Kokkoris for implementing orthorectification. We would finally like to thank the Editor and two anonymous Reviewers for providing feedback that improved the manuscript.

**Conflicts of Interest:** The authors declare no conflict of interest.

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
