# Peer review of "Simulating Future Land Use and Cover of a Mediterranean Mountainous Area: The Effect of Socioeconomic Demands and Climatic Changes"

_land, doi:10.3390/land12010253_

Round 1
Reviewer 1 Report
Summary
The authors applied a dynamic land use and land cover change model to predict a land use/land cover situation for 2055 for an area in Greece, representing a typical southern European sub-mountainous area. In total four demand scenarios, each comparing three climatic scenarios, were assumed. In addition, the spatial distribution of LUC among the different scenarios was investigated. In general, a decrease of farmland and an increase of forests is expected. The scenarios also outlined a shift of forests to higher altitudes, whereas grassland and shrubs will shift until 2055 to lower areas. Validations to proof the models were carried out.
General Comments
In general, politicians need information about the current situation of land for decision making. They also need information about the future impacts of endogen factors on land, e.g. climate change. Simulations are suitable tools to estimate the future situation based on past time series and assumptions. In this way, the current article is very important to show up the possibilities of simulating a future land cover / land use by assuming different scenarios. In addition, the topic of the article fits to the aims of the journal.
The structure of the paper is according to a scientific publication. The introduction gives a good insight into the topic being investigated. The applied methods are described properly, mainly in a narrative form. The results of the investigations are documented and visualized in numerous figures. In chapter 4, the results are discussed and compared with findings of other studies.
The paper is very technical due to the numerous of methods applied for the projections of the numerous input parameters and for the simulations of different scenarios. This is a challenge when reading the paper.
For better understanding, the identifier of the individual scenarios should be reconsidered: The numbers SSPx could be replaced by abbreviations corresponding to the scenarios.
In general, figures documented in the article are very informative. Only figure 5 and figure 8 have potential for improvement, as both can only be understood with the extensive description placed in the label.
The authors justify the choice of the year 2055 for the prediction of land use with the data available for their study. At least in the conclusions they can point out for which other time periods their model would be useful. As the results documented in the study are strongly dependent on the chosen input data and methods, from the reviewer's point of view, potential improvements of the presented approach could be indicated.
Detailed suggestions for improvements
Line 34: Add “Simulation” to the keywords
Line 160: Give information about the accuracy of LUC-mapping
Line 186ff: Add a short description of the scenarios SSP126, SSP370 and SSP585 for better understanding. Explain CMIP6 framework. Use acronyms related to the scenarios (e.g. SSP-SUS)
Line 190: Explain CHELSAfuture V2.1 dataset and CHELSAcruts (Line 200)
Line 333: Explain ROC curve and AUC
Line 464: Add a table including the results (including size of area and %)
Author Response
Dear Editor and Reviewer,
We hereby append the responses to your comments for our manuscript entitled “Simulating future land use and cover of a Mediterranean mountainous area: the effect of socioeconomic demands and climatic changes”. Thank you very much for the valuable review of the manuscript!
We refer to your comments with capital C, followed by the code number of the Editor (number 0) and the Reviewer (number 1 or 2), followed by the order of the comment as it appeared in the review. For example, the comments of Reviewer 1 are coded as C.1.1, C.1.2, ..., etc. Correspondingly, our replies adopt the same coding scheme, but the code starts with capital R instead of C, e.g. R.1.3 is the response to the 3rd comment of Reviewer 1. Where applicable, we identified in-text comments, and tagged them with curly brackets therein. Our comments about content in specific lines of the manuscript’s previous submission are indicated by capital "L" followed by the line numbers, e.g. "L23-25" points to lines 23 to 25 in the old version. Lines from the new version of the manuscript are indicated by a leading “n”, e.g. "nL23-25" points to lines 23 to 25 in the revised manuscript.
Best regards,
the authors of land-2045795
~~~~~~~~~~~~~~~~~~~~~~~~~~~~~~~~~~~~~~
Editor’s comments
Introduction:
Too much methodological detail about the applied model in introduction section. I suggest to move lines 66-90 to the methods section. {C.0.1}
R.0.1: Indeed, this was too technical content for an Introduction, thank you. We have now moved L63-83 to the Methods section (nL169-191).
On the other hand, there is not enough background on the motivation of the study in the introduction {C.0.2}. Why is this study needed? Which is/are the explicit research gap(s) the study closes? Why is it a challenging terrain {C.0.3}?
R.0.2: We have now attempted to cover these concerns by adding two relevant paragraphs in the Introduction (nL50-65).
R.0.3: We wrote and submitted this manuscript after an invitation we had from the Section Managing Editor Ms. Irelia Wang to submit a manuscript to the Journal. At the stage of submission, Ms. Wang with whom we were in touch suggested to us, based only on the title and abstract of the manuscript, to submit it to this Special Issue. If you consider that our manuscript does not fit in the context of the Special Issue, we would like to be opt out from it, and our submission to be included in the regular Journal's issues. Actually, our study has not been developed under the topic of challenging terrains.
Futher, the relevance onf this study to climate mitigation is not introduced in the introduction, but only in the discussion/conclusion section. {C.0.4}
R.0.4: This part is now covered by the addition of a relevant paragraph in the Introduction (nL50-57).
Methods:
I suggest to start with introducing the study area. {C.0.5}
R.0.5: Indeed, that was a helpful suggestion, and we have now implemented it (nL75).
I suggest to name give explicit names to the demand scenarios in the methods section (as displayed in Figure 3). {C.0.6}
R.0.6: We have now added the explicit names as in Figure 3 in the relevant subsection of the Methods (nL192-220).
~~~~~~~~~~~~~~~~~~~~~~~~~~~~~~~~~~~~~~
Reviewer 1 comments
Summary
The authors applied a dynamic land use and land cover change model to predict a land use/land cover situation for 2055 for an area in Greece, representing a typical southern European sub-mountainous area. In total four demand scenarios, each comparing three climatic scenarios, were assumed. In addition, the spatial distribution of LUC among the different scenarios was investigated. In general, a decrease of farmland and an increase of forests is expected. The scenarios also outlined a shift of forests to higher altitudes, whereas grassland and shrubs will shift until 2055 to lower areas. Validations to proof the models were carried out.
General Comments
In general, politicians need information about the current situation of land for decision making. They also need information about the future impacts of endogen factors on land, e.g. climate change. Simulations are suitable tools to estimate the future situation based on past time series and assumptions. In this way, the current article is very important to show up the possibilities of simulating a future land cover / land use by assuming different scenarios. In addition, the topic of the article fits to the aims of the journal.
The structure of the paper is according to a scientific publication. The introduction gives a good insight into the topic being investigated. The applied methods are described properly, mainly in a narrative form. The results of the investigations are documented and visualized in numerous figures. In chapter 4, the results are discussed and compared with findings of other studies.
The paper is very technical due to the numerous of methods applied for the projections of the numerous input parameters and for the simulations of different scenarios. This is a challenge when reading the paper.
For better understanding, the identifier of the individual scenarios should be reconsidered: The numbers SSPx could be replaced by abbreviations corresponding to the scenarios. {C.1.1}
R.1.1: Indeed, this suggestion facilitates the reading of the scenarios. We have now changed throughout the naming scheme to SSP-SUST, SSP-RIVAL and SSP-FUEL.
In general, figures documented in the article are very informative. Only figure 5 and figure 8 have potential for improvement, as both can only be understood with the extensive description placed in the label. {C.1.2}
R.1.2: Indeed, Figs. 5 and 8 are relatively more difficult read, but we believe that the extensive captions facilitate their understanding.
The authors justify the choice of the year 2055 for the prediction of land use with the data available for their study. At least in the conclusions they can point out for which other time periods their model would be useful {C.1.3}. As the results documented in the study are strongly dependent on the chosen input data and methods, from the reviewer's point of view, potential improvements of the presented approach could be indicated. {C.1.4}
R.1.3: We have now added a couple of sentences in the Conclusions, discussing the possibility of investigating further in the future (nL540-543).
R.1.4: We have also added a paragraph in the Conclusions about potential improvements of our current approach (nL543-547).
Detailed suggestions for improvements
Line 34: Add “Simulation” to the keywords {C.1.5}
R.1.5: Thank you for the suggestion; done.
Line 160: Give information about the accuracy of LUC-mapping {C.1.6}
R.1.6: We had dedicated a paragraph in the Methods for the validation of the LUC mapping, and hence for its accuracy (nL282-295). We can understand that this might be skipped by the readers, and hence we added a note in the Results as well (nL373).
Line 186ff: Add a short description of the scenarios SSP126, SSP370 and SSP585 for better understanding. Explain CMIP6 framework. Use acronyms related to the scenarios (e.g. SSP-SUS) {C.1.7}
R.1.7: We have now added a description of CMIP6, and of each scenario (nL105-111).
Line 190: Explain CHELSAfuture V2.1 dataset and CHELSAcruts (Line 200) {C.1.8}
R.1.8: We have now added short explanations of the two datasets (nL114,119-121).
Line 333: Explain ROC curve and AUC {C.1.9}
R.1.9: We have now added explanation for ROC and AUC (nL243-247).
Line 464: Add a table including the results (including size of area and %) {C.1.10}
R.1.10: We suppose that the Reviewer refers to Fig. 3. We believe that a table hosting both the absolute and the relative cover of five land types and for the six demand scenarios would be too cumbersome to read. Figure 3 instead provides a graphical summary of these results, with important for the narrative numbers given in-text (nL324-346).
~~~~~~~~~~~~~~~~~~~~~~~~~~~~~~~~~~~~~~
Author Response
Dear Editor and Reviewers,
We hereby append the responses to your comments for our manuscript entitled “Simulating future land use and cover of a Mediterranean mountainous area: the effect of socioeconomic demands and climatic changes”. Thank you very much for the valuable review of the manuscript!
We refer to your comments with capital C, followed by the code number of the Editor (number 0) and the Reviewer (number 1 or 2), followed by the order of the comment as it appeared in the review. For example, the comments of Reviewer 1 are coded as C.1.1, C.1.2, ..., etc. Correspondingly, our replies adopt the same coding scheme, but the code starts with capital R instead of C, e.g. R.1.3 is the response to the 3rd comment of Reviewer 1. Where applicable, we identified in-text comments, and tagged them with curly brackets therein. Our comments about content in specific lines of the manuscript’s previous submission are indicated by capital "L" followed by the line numbers, e.g. "L23-25" points to lines 23 to 25 in the old version. Lines from the new version of the manuscript are indicated by a leading “n”, e.g. "nL23-25" points to lines 23 to 25 in the revised manuscript.
Best regards,
the authors of land-2045795
~~~~~~~~~~~~~~~~~~~~~~~~~~~~~~~~~~~~~~
Editor’s comments
Introduction:
Too much methodological detail about the applied model in introduction section. I suggest to move lines 66-90 to the methods section. {C.0.1}
R.0.1: Indeed, this was too technical content for an Introduction, thank you. We have now moved L63-83 to the Methods section (nL169-191).
On the other hand, there is not enough background on the motivation of the study in the introduction {C.0.2}. Why is this study needed? Which is/are the explicit research gap(s) the study closes? Why is it a challenging terrain {C.0.3}?
R.0.2: We have now attempted to cover these concerns by adding two relevant paragraphs in the Introduction (nL50-65).
R.0.3: We wrote and submitted this manuscript after an invitation we had from the Section Managing Editor Ms. Irelia Wang to submit a manuscript to the Journal. At the stage of submission, Ms. Wang with whom we were in touch suggested to us, based only on the title and abstract of the manuscript, to submit it to this Special Issue. If you consider that our manuscript does not fit in the context of the Special Issue, we would like to be opt out from it, and our submission to be included in the regular Journal's issues. Actually, our study has not been developed under the topic of challenging terrains.
Futher, the relevance onf this study to climate mitigation is not introduced in the introduction, but only in the discussion/conclusion section. {C.0.4}
R.0.4: This part is now covered by the addition of a relevant paragraph in the Introduction (nL50-57).
Methods:
I suggest to start with introducing the study area. {C.0.5}
R.0.5: Indeed, that was a helpful suggestion, and we have now implemented it (nL75).
I suggest to name give explicit names to the demand scenarios in the methods section (as displayed in Figure 3). {C.0.6}
R.0.6: We have now added the explicit names as in Figure 3 in the relevant subsection of the Methods (nL192-220).
~~~~~~~~~~~~~~~~~~~~~~~~~~~~~~~~~~~~~~
Reviewer 2 comments
In the paper “Simulating future land use and cover of a Mediterranean mountainous area: the effect of socioeconomic demands and climatic changes”, the authors describe a land use-land use change (LULUC) model and how they use the model to project future land use in a mountainous region of Northern Greece. They track the ratio of forest, scrub, grassland and farmland and observe how the ratios may change under four different socioeconomic and three different climate scenarios. The paper is interesting to read and provides much detail on the methods. However, due to the flood of information, the main message gets lost a little and should be stated more clearly {C.2.1}. The authors should also critically read the paper again and edit it for brevity and clarity {C.2.2}. Throughout the paper, the authors should also check the use of the abbreviation LUC (land use change). Sometimes it is correct, but more often than not the authors seem to be actually talking about LU (land use), not LUC (land use change). For example lines 98, 100, 118. {C.2.3}
R.2.1: We have tried to improve the clarity of the manuscript, which we believe is also increased with the addition of the two motivation paragraphs in the Introduction.
R.2.2: We have tried to improve the brevity of the manuscript as well.
R.2.3: We think that the Reviewer missed that in our work “LUC” refers to Land Use and Cover, and not Land Use Change. Consequently, we use “LUC change” in the places we indeed refer to changes in LUC.
Introduction
There is too much methodological detail about the applied model in introduction section. I suggest merging the information in lines 81-114 with the model description in the methods section {C.2.4}.
R.2.4: We have now moved the more technical parts of the Introduction to the Methods (nL169-191).
There is not enough background on the motivation of the study in the introduction {C.2.5}. Why is this study needed? Which is/are the explicit research gap(s) the study closes? Is this a continuation of the fieldwork the authors did in the area, where they use a model to project their findings into the future? Or is it a methodological paper that presents the methodology for the first time? Please clarify. The authors should also introduce the factor mitigation potential of forests in this section {C.2.6}. They mention the aspect in the discussion, but should introduce it here.
R.2.5: We have now added two paragraphs for the motivation of the study (nL50-65).
R.2.6: We have now added related content in the Introduction (nL63).
Materials and Methods
I commend the authors in providing this much detail, but overall, I was quite overwhelmed by this section. I had trouble identifying the information that was most relevant for me in order to be able to interpret the results. My recommendation would be to edit the section for clarity {C.2.7}, and maybe move some of the information only relevant for scientists looking to replicate the study to an appendix {C.2.8}.
R.2.7: We have tried to structure the Methods with an extra nested level of sub-headings, to make information more easy to access.
R.2.8: We’d prefer to have all methodological content in the main text, as the journal Land suggests in the author guidelines.
I suggest starting this section with section 2.1., not with Fig. 1 {C.2.9}
R.2.9: We restructured the Methods, to start with the description of the study area (nL75). Previous Fig. 1 has now become Fig. 2.
Section 2.3: You speak a lot of demand in this section. To me (and I assume many others) demand is primarily an economic term denoting the demand for a product. What exactly is the demand for here? A certain land use type? A change in land use type? Demand for agricultural/forest products? {C.2.10}
R.2.10: “Demand”, in the sense we commonly use it in the manuscript, is the technical term used in LUC change models to denote the demand in land type cover or transitions in the future map to be predicted by the models. We have now added a clarification in the first mention of the term in the Methods (nL193).
Lines 267-294: Please label each of your four scenarios with a descriptive name (as done in Fig. 3) {C.2.11}
R.2.11: We have now labeled the introductory parts of the demand scenarios, to make them clearer (nL195-220).
Line 294-297: I have trouble understanding what this scenario entails. Please elaborate more clearly/in more detail {C.2.12}
R.2.12: We have now added more information for this scenario (nL214-220).
Lines 383-386: Do I understand correctly that in theory a cell can change from forest to grassland to forest again in the span of 2 years? Should there not be a lag-time for the formation of a forest cell via scrub cell? {C.2.13}
R.2.13: In our study there were no 2-year intervals. The most simple model, could in theory allow such changes, but if such changes do not appear in the historical transition matrix of the latest time interval, then such changes would be highly improbable to occur in a future time interval of equal length, even with the simplest model of no extra settings forbidding specific changes.
Results
Line 452-453: I don’t understand why only the no demand scenario can vary between climate scenarios. Please elaborate. {C.2.14}
R.2.14: We have now added the explanation (nL325).
Figs. 3, 4, 6, 8: I strongly suggest choosing different colors for the different land use types. The shades of green are very hard to distinguish. {C.2.15}
R.2.15: We understand the concern of the Reviewer, but we decided to keep this colour scheme, because it’s the same in the previous two studies regarding LUC in this study area.
Fig 5: I believe this information would be better presented in a Table {C.2.16}
R.2.16: We decided to use a figure for this information, because there are also the lower and upper theoretical boundaries, and hence all the numbers are more easily read and comparable between demand scenarios.
Lines 592-593: The location of farmland may be more remote in the future period, but I doubt that remoteness is the deciding factor, because that would not make sense from an economic point of view. {C.2.17}
R.2.17: Indeed, we agree with that. As we note in the Discussion, the location of farmland under warmer and drier climate is more related to higher elevation where distances to nearest settlements are greater.
Discussion:
There is a lot of repetition of results in the discussion {C.2.18}. The authors should focus more on providing an interpretation and on expounding the ramifications of their findings on land use planning and mitigation potential {C.2.19}. They should also mention if different or the same trends have been projected for similar ecosystems in other regions. {C.2.20}
R.2.18: Because of the plethora of methodologies and results, we believe that some summary of the results in the Discussion are important reminder for the interpretation that is going to take place.
R.2.19: We believe that focusing more on planning and mitigation measures would concern a paper with a more applied approach, which would perhaps require the inclusion of other parameters for the design of management measures. In other words, focusing more on the applied side, our arguments would sound even more unsupported without actual evidence from the study to support them. Our manuscript is more focused on methodology development for predicting and highlighting relations between drivers and LUC change.
R.2.20: Since the topic of the manuscript is the Mediterranean, we would prefer to not expand to other regions, because the Discussion would be even more cumbersome to read.
Line 612: “The” is missing at the beginning of the sentence {C.2.21}
R.2.21: Added.
Lines 626-645: Much of the information is superfluous and can be shortened to the following information: “In our climate scenarios, the expected warming rates were +1.8°C and +3°C, and the estimated change in annual precipitation was -3.1% and -8.9% under SSP126 and SSP585, respectively.” {C.2.22}
R.2.22: We prefer to compare temperature in a dedicated sentence, and precipitation in a separate one. This is the reason we preferred this structure. Might be more superfluous, but it eases the reading and emphasises the comparison.
Lines 707-763 are almost exclusively repeating information already given in the methods and results section. Please extract the relevant information and discuss it in relation to findings of other studies. {C.2.23}
R.2.23: We believe that this detailed explanation is important for understanding why the results came up the way they did, especially in a complex model like the present one. We have attempted to discuss our findings in relation to other studies (nL520-526).